# Testing the Impact of Exports, Imports, and Trade Openness on Economic Growth in Namibia: Assessment Using the ARDL Cointegration Method

**Tafirenyika Sunde [1,2,*], Blessing Tafirenyika [3] and Anthony Adeyanju [3]**

[1] Trade and Development Department, Faculty of Economic and Management Sciences, Potchefstroom Campus, North-West University, Potchefstroom 2520, South Africa

[2] Department of Economics, Accounting and Finance, Faculty of Commerce, Human Sciences and Education, Namibia University of Science and Technology, Windhoek 9000, Namibia

[3] Department of Economics and Mathematics, Faculty of Business Administration and Strategic Management, International University of Management, Windhoek 9000, Namibia

[*] Correspondence: tsunde@nust.na

**Abstract:** This study examines the impact of exports, imports, and trade openness on Namibia's economic growth using the ARDL cointegration method. The results reveal a significant negative relationship between imports and economic growth, while exports and trade openness show positive and significant relationships with economic growth. Moreover, short-term economic growth is driven by exports, imports, and trade openness. The findings suggest that trade liberalisation and export-led growth are crucial for Namibia's economic development. Overall, this study supports the mercantilist theory, which emphasises the importance of participating in global markets by increasing exports and trade.

**Keywords:** economic growth; exports; imports; trade openness; export-led growth; ARDL; unit roots; cointegration; ECM

## 1. Introduction

The relationship between trade and economic growth has been the subject of debate among scholars, with two distinct schools of thought emerging. The first is the export-led growth strategy, which views trade as the engine of growth (Riedel 1984; Iqbal et al. 2019), while the second is import substitution, which regards trade as "the servant of growth" (Kravis 1970; Michaely 1977). Empirical evidence on the relationship between trade and economic growth remains divided, but studies by (Awokuse 2003; Shirazi and Manap 2005; Dreger and Herzer 2013) have confirmed the export-led growth hypothesis. This indicates that strong export sectors can benefit from increased resource allocation, capacity utilisation, scale economies, and technological innovation spurred by global market competition (Helpman and Krugman 1989). However, the importance of imports in driving economic expansion should not be overlooked (Bakari and Mabrouki 2017). Imports can provide domestic firms access to foreign technology and intermediate inputs, promoting long-term economic growth.

It must be noted that very little research has been conducted in Namibia on the relationship between trade and economic growth. The few researchers who investigated the link between international trade and economic expansion in Namibia include Jordaan and Eita (2007) and Mosikari and Eita (2020), who studied the link between exports and economic growth. The current study explores how exports, imports, and trade openness impact economic growth in Namibia using an ARDL multivariate framework. Imports and exports show the importance of trade to Namibia's economic growth, whereas trade openness shows the importance of trade liberalisation.

Our study augments the existing body of knowledge on the effects of exports, imports, and trade openness on economic growth in the ways described below. First, although numerous studies on the impact of imports and exports on economic growth have been undertaken in various countries, there have been very few country-specific studies on small African states such as Namibia. Second, prior research used cross-country data to build panel models that ignored country-specific traits and eccentricities (see Sanjuán-López and Dawson 2010; Mehrara and Baghbanpour 2016; Hamdan 2016; Meyer 2021). Additionally, when nations at different stages of development are combined, the panel data approach fails to account for the country-specific effects of exports and imports on economic growth. Furthermore, when nations are assessed collectively, positive and negative effects may cancel the other out, making it impossible to discern how these variables interact in each nation. Third, several early studies neglected long-term econometric relationships, which is a key source of problems regarding the validity of the findings. In line with this, researchers face challenges because of their econometric methodologies and the scarcity of available data. The majority of previous studies used vector autoregression (VAR) to examine the bivariate relationship between exports and economic growth (see Shan and Tian 1998; Chuang 2000; Awokuse 2005; Ajmi et al. 2015). This method has significant drawbacks when the sample size is small, as is the case in the majority of developing countries. Fourth, variable omission bias influences bivariate exports and economic growth models, as well as imports and economic growth (see Al-Yousif 1999; Amoateng and Amoako-Adu 1996). Economic growth in such a model can only be explained by the lags of itself and the lags of the other variable. To address the issue of variable omission bias in previous studies, the current analysis incorporates additional variables that explain economic growth, such as capital, general government final consumption expenditure, and inflation proxy (GDP deflator). Finally, the ARDL approach employed in this study is more adaptable than the VAR methodology used in many previous studies, which requires all variables to be integrated of the same order. Because it can be used when all variables are integrated of order 0, integrated of order 1, or a combination of both, the ARDL approach is more adaptable.

The rest of the article is divided into the following sections. Section 2 provides a synopsis of the empirical literature, while Section 3 discusses the empirical data and methodology. Section 4 provides the results and discussion, and Section 5 gives the conclusion and recommendations of the study.

## 2. Empirical Literature Review

Theoretical and empirical research on the relationships between exports, imports, trade openness, and economic growth is extensive. Traditional trade theory explains trade's static advantages due to competition and specialisation based on comparative advantage. Even after accounting for these benefits in terms of national output, they can still positively affect economic growth as economies readjust to a new equilibrium due to international trade liberalisation, which increases access to global markets.

Numerous theoretical models provide insights into the dynamic benefits of international trade, following in the footsteps of "endogenous growth theories" pioneered by Romer (1986) and Pack (1994). Helpman and Krugman (1989) and Grossman and Helpman (1991), Romer (1994), Barro and Sala-I-Martin (1995), Coe and Helpman (1995) all argue that a country's openness to trade influences technological change, which in turn influences economic growth. Several empirical studies have been conducted to investigate the relationship between international trade and economic growth, and the findings are frequently inconsistent and contradictory across methodologies and countries. Trade, according to Freund and Bolaky (2008), Marelli and Signorelli (2011), Chirwa and Odhiambo (2016) and Frankel and Romer (2017), stimulate economic growth. However, according to Vamvakidis (2002), Rigobon and Rodrik (2005), and Ulaşan (2015), international trade has an adverse impact on economic growth.

In addition to theoretical advances in the trade and growth literature, a growing body of empirical literature has attempted to test the export-led growth hypothesis using various methods and data sets. Early empirical research, such as that conducted by Michaely (1977) and Tyler (1981), used cross-sectional data from various countries to examine the relationship between exports and economic growth and found that exports affect economic growth. Other recent studies, such as those conducted by Ajmi et al. (2015), Sunde (2017), Yaya (2017), Adebayo (2020), and Tivatyi et al. (2022) also support the export-led growth hypothesis.

When examining the relationship between international trade and economic growth, Riezman et al. (1996) emphasised the importance of imports. According to their study, excluding imports from the analysis could obscure or exaggerate the effects of exports on economic growth. Only 30 of the 126 countries they studied showed unidirectional causality from exports to economic growth when imports were considered. They claimed their findings were more reliable than previous studies, which have ignored the role of imports in the trade and economic growth relationship. Other authors, including Saaed and Hussain (2015) and Bakari and Mabrouki (2017), have also acknowledged the role and importance of imports in the relationship between trade and economic growth. The findings of these studies show a long-run equilibrium relationship between exports, imports, and output growth.

Numerous empirical studies on the relationship between trade openness and economic growth have been conducted; however, the findings of these studies are frequently inconsistent and contradictory across methodologies and countries. According to Rassekh (2007), low-income economies benefit more from international trade than high-income economies. In a study of 82 industrialised and developing countries, Chang et al. (2009) discovered a strong positive association between trade openness–economic growth relationships. Kim and Lin (2009) studied 61 countries and discovered an income threshold above which increased trade leads to increased economic growth. They discovered that trade openness stifles economic growth below a certain threshold. In a similar study, Musila and Yiheyis (2015) discovered that increased economic growth in Kenya results from increased investment rather than trade liberalisation. Furthermore, Lawal et al. (2016) investigated trade openness in Nigeria using ARDL and discovered that it has a negative long-term effect but a positive short-term effect on economic growth.

Kim et al. (2011) investigated the relationship between trade openness and economic growth using instrumental variable threshold regressions. They first demonstrated that trade openness promotes economic growth, financial development, and capital accumulation in high-income countries while it inhibits economic growth in low-income countries. Second, they discovered that inflation and financial growth affect international trade. Third, they also discovered that trade openness stifles economic growth in low-income countries without affecting high-income countries. Finally, they discovered that trade openness boosts economic growth in countries with low inflation but not in countries with high inflation. Dufrenot et al. (2010) used quantile regression to examine the trade openness–growth nexus for 75 developing countries in a similar study. They concluded that trade openness benefits low-growth countries more than high-growth countries.

Several studies also investigated the trade openness–economic growth relationship in Sub-Saharan Africa. First, Tekin (2012) discovered no link between foreign aid, trade openness, and per capita GDP in 27 African LDCs. Second, Asfaw (2014) investigated the impact of trade liberalisation on 47 Sub-Saharan African countries and concluded that trade liberalisation promotes economic growth and investment and that trade policy, such as the average weighted tariff rate and the real effective exchange rate, affects economic performance. Third, Menyah et al. (2014) investigated the interactions between trade openness, financial development, and economic growth in twenty-one Sub-Saharan African countries. They concluded that the theory of trade openness driving economic growth is unconvincing. Trade openness, they discovered, drives economic growth in Benin, Sierra Leone, and South Africa. Fourth, Brueckner and Lederman (2015) examined 41

Sub-Saharan African countries using instrumental variables and concluded that trade liberalisation benefits economic growth in the short and long term.

The relationship between Namibia's trade and economic growth has received little attention. Jordaan and Eita (2007) examined the relationship between exports and GDP in Namibia between 1970 and 2005 in a separate study. They used Granger causality and cointegration methodology to show that exports contribute significantly to economic growth; thus, the study supports an export-led growth strategy. Mosikari and Eita (2020) investigated Namibia's major export sectors (manufactured goods, diamond mining, food, and live animals) and economic growth. Quarterly data from 2009 to 2018 were subjected to a nonlinear autoregressive distributive lag. This study discovered an asymmetrical relationship between Namibia's most important export sectors and economic growth.

## 3. Data and Methodology

In this study, the ADF and PP Tests for unit roots, as well as the autoregressive distributive lag model cointegration approach, are used. The ARDL approach is comprised of the Wald test, the long-run OLS estimation test, the error correction and short-run relationship estimation test, and the short-run causality test. The World Bank database, Bank of Namibia statistics, and the National Statistical Agency of Namibia database were the three primary sources from which the data for the model variables were derived. The models used in this study were based on data collected from 1990 to 2020.

### 3.1. Model Specification

The estimated equations take the forms represented by Equations (1) and (2) below:

$$GDPPC = \alpha_0 + \alpha_1 CAPITAL + \alpha_2 GEN + \alpha_3 EXPORT + \alpha_4 IMPORT + u_1 \tag{1}$$

$$GDPPC = \beta_0 + \beta_1 CAPITAL + \beta_2 GEN + \beta_3 TROPEN + \beta_4 DEFLATOR + u_2 \tag{2}$$

where
GDPPC = per capita real GDP (a proxy for economic growth).
IMPORT = imports of goods and services as a percentage of GDP.
EXPORT = exports of goods and services as a percentage of GDP.
GNE = general government final consumption expenditure as a percentage of GDP.
CAPITAL = gross fixed capital formation.
TROPEN = trade openness = $\left( \frac{Exports + Imports}{GDP} \times 100 \right)$.
DEFLATOR = measure of inflation.

CAPITAL, EXPORT, GNE, AND TROPEN are expected to affect the proxy for economic growth (GDPPC) positively. In other words, increases in all these variables lead to economic growth. Imports can have both positive and negative effects on economic growth. When the economy imports capital equipment used to produce other goods, this positively affects economic growth when the country commences production. However, imports of goods and services may negatively affect economic growth because imported goods substitute domestic production. Finally, inflation can also positively or negatively impact economic growth. Inflation caused by increased aggregate demand during a boom may positively affect economic growth, while inflation which occurs when the economy performs poorly, negatively affects economic growth.

### 3.2. ARDL Model Specification

The ARDL cointegration technique was first developed by Pesaran et al. (1999) and eventually improved by Pesaran et al. (2001). It has three advantages over earlier and more traditional cointegration techniques. First, the ARDL method does not require all variables to be of the same order of integration; it can be employed when the underlying series are integrated of order one, order zero, or fractionally. The second advantage of the model is that it is more effective in smaller and more limited data sample sizes. The

ability of the ARDL method to generate non-biased long-run model estimates is the third advantage of using this methodology (Harris and Sollis 2003). The ARDL models for the current investigation are represented by Equations (3) and (4) below:

$$
\begin{aligned}
\Delta \text{GDPPC}_t = \ & \rho_0 + \rho_1 \text{GDPPC}_{t-1} + \rho_2 \text{CAPITAL}_{t-1} + \rho_3 \text{GNE}_{t-1} + \rho_4 \text{EXPORT}_{T-1} \\
& + \rho_5 \text{IMPORT}_{t-1} \\
& + \sum_{i=1}^{n} \rho_{11i} \Delta \text{GDPPC}_{t-i} \\
& + \sum_{i=1}^{n} \rho_{12i} \Delta \text{CAPITAL}_{t-i} \\
& + \sum_{i=1}^{n} \rho_{13i} \Delta \text{GNE}_{t-i} + \sum_{i=1}^{n} \rho_{14i} \Delta \text{EXPORT}_{t-i} \\
& + \sum_{i=1}^{n} \rho_{15i} \Delta \text{IMPORT}_{t-i} + \nu_{1t}
\end{aligned} \tag{3}
$$

$$
\begin{aligned}
\Delta \text{GDPPC}_t = \ & \sigma_0 + \sigma_1 \text{GDPPC}_{t-1} + \sigma_2 \text{CAPITAL}_{t-1} + \sigma_3 \text{GNE}_{t-1} + \sigma_4 \text{TROPEN}_{T-1} \\
& + \sigma_5 \text{DEFLATOR}_{t-1} \\
& + \sum_{i=1}^{n} \sigma_{11i} \Delta \text{GDPPC}_{t-i} \\
& + \sum_{i=1}^{n} \sigma_{12i} \Delta \text{CAPITAL}_{t-i} \\
& + \sum_{i=1}^{n} \sigma_{13i} \Delta \text{GNE}_{t-i} \\
& + \sum_{i=1}^{n} \sigma_{14i} \Delta \text{TROPEN}_{t-i} + \sum_{i=1}^{n} \sigma_{15i} \text{DEFLATOR}_{t-i} + \nu_{2t}
\end{aligned} \tag{4}
$$

where

$\rho_1 - \rho_5$ and $\sigma_1 - \sigma_5$ denote the long-run multipliers.
$\rho_0$ and $\sigma_0$ are the intercepts.
$\rho_{11} - \rho_{15}$ and $\sigma_{11} - \sigma_{15}$ are short-run dynamic coefficients.
$\mu_{1t}$ and $\mu_{2t}$ are the error terms.

### 3.3. ARDL Bounds Test Decision Rules

For each level of significance, two different sets of critical values are available. We assume that all variables are integrated of order zero [I(0)] when calculating the first critical value and assume that all variables are integrated of order one [I(1)] when calculating the second critical value. When the test statistic exceeds the upper critical bounds value, the null hypothesis of no cointegration is rejected. The null hypothesis is considered true when the F-statistic is less than the value of the lower critical bound. If the variables in the models are cointegrated, the ARDL error correction method, as shown in Equations (5) and (6), can be used to find the long-run coefficients.

$$
\begin{aligned}
\Delta \text{GDPPC}_t = \ & \varphi_1 + \sum_{i=1}^{n} \theta_{1i} \Delta \text{GDPPC}_{t-i} + \sum_{i=1}^{n} \theta_{2i} \Delta \text{CAPITAL}_{t-i} + \sum_{i=1}^{n} \theta_{4i} \Delta \text{GNE}_{t-i} \\
& + \sum_{i=1}^{n} \theta_{3i} \Delta \text{EXPORT}_{t-i} + \sum_{i=1}^{n} \theta_{4i} \Delta \text{IMPORT}_{t-i} + \lambda_1 \text{ECT}_{1t-1} + \mu_{1t}
\end{aligned} \tag{5}
$$

$$
\begin{aligned}
\Delta \text{GDPPC}_t = \ & \varphi_2 + \sum_{i=1}^{n} \gamma_{1i} \Delta \text{GDPPC}_{t-i} + \sum_{i=1}^{n} \gamma_{2i} \Delta \text{CAPITAL}_{t-i} + \sum_{i=1}^{n} \gamma_{4i} \Delta \text{GNE}_{t-i} \\
& + \sum_{i=1}^{n} \gamma_{3i} \Delta \text{TROPEN}_{t-i} + \sum_{i=1}^{n} \gamma_{4i} \Delta \text{DEFLATOR}_{t-i} \\
& + \lambda_2 \text{ECT}_{2t-1} + \mu_{2t}
\end{aligned} \tag{6}
$$

where $\text{ECT}_{t-i}$ are the error correction terms and $\lambda_1$ and $\lambda_2$ are the coefficients of the error correction terms in Equations (5) and (6), respectively. All the variables are as defined previously.

## 4. Results

### 4.1. The Unit Root Tests

Even though the tests for unit roots are unnecessary when using ARDL methodology, we tested for unit roots to ensure that no variable was integrated of order 2. We employed the Augmented Dickey–Fuller (ADF) and the Phillips–Perron (PP) tests to test for unit roots. The unit root test results for this study are shown in Table 1 below.

**Table 1.** Unit root test results.

| Variable | PP | | ADF | | Decision |
|---|---|---|---|---|---|
| | **Levels** | **1st Difference** | **Levels** | **1st Difference** | **I(d)** |
| GDPPC | −1.067856 | −2.895649 * | −0.637260 | −2.895649 * | I(1) |
| CAPITAL | −0.740966 | −4.38133 *** | −1.003957 | −4.4612 *** | I(1) |
| GNE | −2.666170 | −4.81353 *** | −2.525702 | −4.8663 *** | I(1) |
| EXPORT | −0.867513 | −4.96835 *** | −0.867513 | −4.8946 *** | I(1) |
| IMPORT | −0.466587 | −5.99031 *** | −2.566135 | −6.0111 *** | I(1) |
| TROPEN | −0.433793 | −4.94327 *** | −0.333829 | −4.7301 *** | I(1) |
| DEFLATOR | −6.107 *** | | −21.938 *** | | I(0) |

Note: *** and * signify stationarity at the 1% and 10% significance levels, respectively. Source: Authors' compilation.

Some variables have order one integration [I(1)], while others have order zero integration [I(0)]. Unlike other techniques, the ARDL cointegration method does not require unit root pre-tests. As a result, when series are of varying orders of integration (such as a combination of I(0), I(1)), the ARDL cointegration method should be used. It is considered robust when only one long-run relationship exists between the underlying variables in a small sample size. Based on the results in Table 1, the ARDL cointegration technique proposed by Pesaran et al. (2001) was chosen as the best estimation method for this study. The primary advantage of using this method is its ability to identify cointegrating vectors in the presence of multiple cointegrating vectors.

### 4.2. The Bounds Test Results

The first phase in the ARDL model's analytical approach is cointegration testing. This test can be carried out using the Wald test, which is meant to test the "null hypothesis" that cointegration does not exist. The bounds tests rely on the joint F-statistic, which has a non-standard asymptotic distribution under the null hypothesis of no cointegration. The first part of the ARDL bounds approach is to estimate the equations using the ordinary least-squares (OLS) method. The F-test is then used to determine whether the variables have a long-term relationship. We investigate the significance of the following hypothesised joint relationships between the coefficients of the lagged levels of the variables by using the following null hypotheses: $H_0$: $\rho_1 = \rho_2 = \rho_3 = \rho_4 = \rho_5 = 0$; and $H_0$: $\sigma_1 = \sigma_2 = \sigma_3 = \sigma_4 = \sigma_5 = 0$. The first null hypothesis is related to Equation (3), whereas the second is related to Equation (4). Table 2 shows the results of the bounds test for cointegration.

**Table 2.** Bounds Test.

| | **Model 1** | | **Model 2** | |
|---|---|---|---|---|
| F-statistic | 8.6670 | | 3.9423 | |
| **Asymptotic** | **I0 bound** | **I1 bound** | **I0 bound** | **I1 bound** |
| 10% | 2.260 | 3.350 | 2.080 | 3.000 |
| 5% | 2.620 | 3.790 | 2.390 | 3.380 |
| 1% | 3.410 | 4.680 | 3.060 | 4.150 |

Source: Authors' compilation.

### 4.3. Estimation of the Long-Run Relationships

Our investigation commences with the estimation of long-run coefficients, followed by the estimation and analysis of error correction models (short-run coefficients). Given that the variables in both models are cointegrated (see Table 2), we do not present the complete set of long-term results. In this instance, the significance of the variables in these long-run models is irrelevant, as the results are erroneous. In light of this, we only present the coefficients and standard errors of both models' long-run results, as shown in Table 3. Long-run results from Model 1 indicate that capital (CAPITAL), gross national expenditure (GNE), and exports (EXPORT) positively affect economic growth (GDPPC), whereas import (IMPORT) negatively affects it. Similarly, the long-run results of Model 2 reveal that capital (CAPITAL), gross national expenditure (GNE), and trade openness (TOPEN) positively affect economic growth (GDPPC), while inflation (DEFLATOR) negatively affects it. In both models, the signs of the coefficients of the independent variables align with the a priori expectations.

**Table 3.** The long-run results. Dependent Variable: GDPPC.

| | Model 1 | | | Model 2 | |
|---|---|---|---|---|---|
| **Variable** | **Coefficient** | **Std. Error** | **Variable** | **Coefficient** | **Std. Error** |
| CAPITAL | 0.285048 | 0.118851 | CAPITAL | 0.394322 | 0.016138 |
| GNE | 0.453270 | 0.035119 | GNE | 0.421142 | 0.159830 |
| EXPORT | 0.749757 | 0.145929 | TOPEN | 0.437013 | 0.148780 |
| IMPORT | −0.270894 | 0.092317 | DEFLATOR | −0.090932 | 0.147759 |
| C | 7.867679 | 0.869250 | C | 11.20406 | 0.644007 |

Source: Authors' compilation.

### 4.4. ECM Results and Analysis

Tables 4 and 5 show the ARDL-ECM results for Models 1 and 2, respectively. The main aim of the results in Table 5 is to test whether exports and imports are important in explaining economic growth in Namibia. First, the results show that capital and its first lag positively impact economic growth in Namibia at the 1 percent significance level. Several empirical studies support this finding in the literature (see Chirwa and Odhiambo 2016; Barro 2003; Kim et al. 2011). Second, gross national expenditure positively influences economic growth in Namibia at the 5% significance level. The second lag of gross national expenditure is insignificant at about 12% significance level. Numerous empirical studies corroborate this finding (see Chirwa and Odhiambo 2016; Barro 2003; Kim et al. 2011). Third, the results show that exports and the first lag of exports positively impact economic growth at the 1% significance level. This finding also has overwhelming support from the empirical literature (see Jordaan and Eita 2007; Ajmi et al. 2015; Sunde 2017; Yaya 2017; Adebayo 2020; Tivatyi et al. 2022; Mosikari and Eita 2020). Fourth, the results also show that the import variable and its first lag negatively impact economic growth in Namibia at the 1% and 5% levels of significance, respectively. This finding is corroborated in the empirical literature by Saaed and Hussain (2015) and Bakari and Mabrouki (2017). Finally, the results also show a long-run economic relationship between economic growth and the independent variables included in Model 1 because the coefficient of the error correction term is negative and significant at the 1% significance level. This result demonstrates that economic growth adjusts towards its long-run equilibrium at a rate of about 9.28% per annum, which implies that full equilibrium will be reached in the 11th year. The existence of a cointegrating relationship between economic growth and independent variables, as well as the importance of exports and imports in explaining economic growth, unequivocally demonstrate the significance of international trade in Namibia in both the short and long run.

**Table 4.** Model 1 ARDL Error Correction Results.

| | Dependent Variable: ΔLNGDPC | | | |
|---|---|---|---|---|
| **Variable** | **Coefficient** | **Std. Error** | **t-Statistic** | **Prob.** |
| $\Delta GDPPC_{t-1}$ | 0.718561 | 0.109843 | 6.541717 | 0.0000 |
| $\Delta CAPITAL$ | 0.023534 | 0.007184 | 3.275855 | 0.0021 |
| $\Delta CAPITAL_{t-1}$ | 0.023481 | 0.007666 | 3.063024 | 0.0038 |
| $\Delta GNE$ | 0.098649 | 0.040458 | 2.438287 | 0.0190 |
| $\Delta GNE_{t-1}$ | 0.064403 | 0.041016 | 1.570206 | 0.1237 |
| $\Delta EXPORT$ | 0.193348 | 0.052318 | 3.695652 | 0.0006 |
| $\Delta EXPORT_{t-1}$ | 0.170057 | 0.049054 | 3.466700 | 0.0012 |
| $\Delta IMPORT$ | −0.107918 | 0.039872 | −2.706626 | 0.0097 |
| $\Delta IMPORT_{t-1}$ | −0.089017 | 0.041593 | −2.140167 | 0.0381 |
| $ECT_{t-1}$ | −0.092793 | 0.018807 | −4.933906 | 0.0000 |
| R-squared | | 0.779004 | | |
| Adjusted R-squared | | 0.707051 | | |
| $\chi^2$ Serial | | 0.345567 (0.7686) | | |
| $\chi^2$ ARCH | | 0.359706 (0.5526) | | |
| $\chi^2$ Normal | | 0.364725 (0.8333) | | |
| $\chi^2$ RESET | | 0.609020 (0.4443) | | |

Source: Authors' compilation.

**Table 5.** Model 2 ARDL Error Correction Results.

| | Dependent Variable: ΔLNGDPC | | | |
|---|---|---|---|---|
| **Variable** | **Coefficient** | **Std. Error** | **t-Statistic** | **Prob.** |
| $\Delta GDPPC_{t-1}$ | 0.753654 | 0.109279 | 6.896593 | 0.0000 |
| $\Delta CAPITAL$ | 0.292955 | 0.100745 | 2.907893 | 0.0056 |
| $\Delta CAPITAL_{t-1}$ | 0.363138 | 0.132785 | 2.734774 | 0.0089 |
| $\Delta GNE$ | 0.025557 | 0.009183 | 2.783107 | 0.0078 |
| $\Delta GNE_{t-1}$ | 0.017339 | 0.007577 | 2.288324 | 0.0269 |
| $\Delta TOPEN$ | 0.041234 | 0.018446 | 2.235314 | 0.0304 |
| $\Delta TOPEN_{t-1}$ | 0.067191 | 0.033958 | 1.978636 | 0.0540 |
| $\Delta DEFLATOR$ | −0.134512 | 0.035039 | −3.838947 | 0.0004 |
| $\Delta DEFLATOR_{t-1}$ | −0.020503 | 0.009262 | −2.213630 | 0.0320 |
| $\Delta DEFLATOR_{t-2}$ | −0.086566 | 0.041356 | −2.093175 | 0.0420 |
| $ECT_{T-1}$ | −0.163381 | 0.025851 | −6.319987 | 0.0000 |
| R-squared | | 0.774172 | | |
| Adjusted R-squared | | 0.744286 | | |
| $\chi^2$ Serial | | 0.249953 (0.7808) | | |
| $\chi^2$ ARCH | | 0.209295 (0.6501) | | |
| $\chi^2$ Normal | | 0.062166 (0.9694) | | |
| $\chi^2$ RESET | | 0.297409 (0.7686) | | |

Source: Authors' compilation.

Model 2 results shown in Table 5 mainly intend to establish the importance of trade openness in explaining economic growth in Namibia. Despite this, we explain all the results we find using this model. First, the results show that capital positively and significantly explains economic growth in Namibia at about 1% significance level. As already demonstrated, several studies support this result in the empirical literature. Second, gross national expenditure and its first lag significantly explain economic growth at the 1% and 5% levels of significance, respectively. This result, as demonstrated earlier, is also supported by several studies in the literature. Third, trade openness and its first lag significantly explain economic growth at a 5% significance level. Several studies support this finding in the literature (see Rassekh 2007; Chang et al. 2009; Kim and Lin 2009; Musila and Yiheyis 2015; Brueckner and Lederman 2015). Fourth, the deflator (a proxy for inflation) significantly explains economic growth in Namibia up to the second lag. Lastly, the coefficient of the

error correction term, which is negative and significant, shows that there is a long-run economic relationship between GDP and the independent variables included in Model 2. The coefficient of the error correction term suggests that economic growth in this model adjusts towards its long-run equilibrium at a rate of 16.33% per annum, which implies that full equilibrium will be reached in the 7th year. The fact that trade openness is significant in explaining economic growth further corroborates the importance of international trade to the Namibian economy.

### 4.5. Diagnostic Tests

The ECM results shown in Tables 4 and 5 pass a battery of diagnostic tests, which are reported in this section. First, the autocorrelation test demonstrates no serial correlation in the models' estimated residuals. Second, the ARCH heteroscedasticity test reveals that the residuals of the estimated models are not heteroscedastic. Third, the findings show that the residuals of the two estimated models follow a normal distribution. Fourth, the Ramsey RESET test results show that both estimated models have valid specifications. Lastly, Figure 1 shows that the CUSUM of squares test confirms parameter stability for both estimated models.

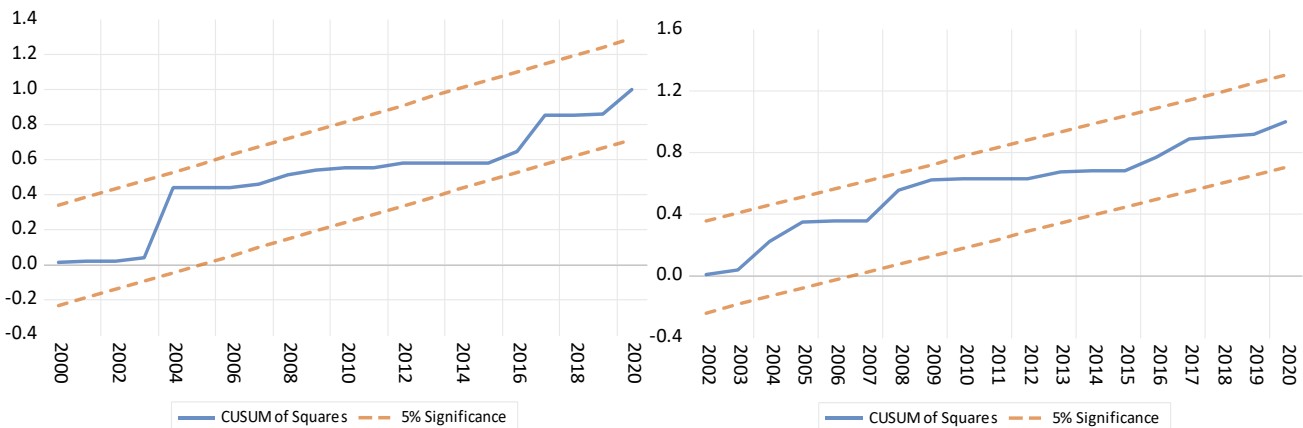

**Figure 1.** Parameter stability tests for Models 1 and 2.

### 5. Conclusions

This article examined the effects of Namibia's exports, imports, and trade openness on economic growth, given that prior empirical research in Namibia had only examined the effects of exports on economic growth. The inclusion of imports in this study was influenced by Riezman et al. (1996), who emphasised the significance of imports in the trade–economic growth relationship and argued that excluding imports from the analysis could obscure or exaggerate the effects of exports on economic growth. In addition, trade openness was included as a factor in the study because trade liberalisation facilitates greater access to international markets, stimulating economic growth.

The research was carried out using the ARDL method. We specified two ARDL economic growth equations. The first equation had exports and imports as some of the independent variables, while the second equation had trade openness as one of the independent variables. According to the findings, exports and trade openness positively and significantly impact Namibia's economic growth. In addition, the study found that imports significantly and negatively affect Namibia's economic growth. The study also found positive and significant effects of capital and general government final consumption expenditure on economic expansion.

Additionally, the study discovered that the deflator (inflation proxy) negatively and significantly impacts Namibia's economic growth. From the preceding statements, the study concludes that international trade and economic growth in Namibia are positively related. Our research findings support the hypothesis that increased global trade stimulates



economic growth. Other researchers, including Freund and Bolaky (2008) and Marelli and Signorelli (2011), have reached similar conclusions. This substantiates mercantilist ideology, which advocates for increased export promotion and Namibia's participation in global markets. Because imports negatively impact Namibia's economic growth, the study recommends that the government can limit imports by imposing quotas and higher import tariffs.

According to two authors who have contributed to research on accelerating economic growth, Galor (2005) and Growiec (2022), the negative effects of international trade on the domestic economy cannot be ignored. They must be mitigated by implementing policies designed to achieve this objective. It would, therefore, be fascinating to examine the negative effects of international trade on Namibia's economy in future studies.

**Author Contributions:** Conceptualisation, T.S., B.T. and A.A.; methodology, T.S., B.T. and A.A.; software, T.S.; validation, B.T. and A.A.; formal analysis, T.S., B.T. and A.A.; investigation, T.S., B.T. and A.A.; resources, B.T.; data curation, B.T.; writing—original draft preparation, A.A.; writing—review and editing, B.T.; visualisation, B.T.; supervision, T.S.; project administration, A.A. All authors have read and agreed to the published version of the manuscript.

**Funding:** This research received no external funding.

**Informed Consent Statement:** Not applicable.

**Data Availability Statement:** Data used and presented in this study are available on request from the corresponding author.

**Conflicts of Interest:** The authors declare no conflict of interest.

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
