# Peer review of "Testing the Impact of Exports, Imports, and Trade Openness on Economic Growth in Namibia: Assessment Using the ARDL Cointegration Method"

_economies, doi:10.3390/economies11030086_

Round 1

Reviewer 1 Report

Table 1 (line 68) - please, check it

"Trade openness shows the importance of trade liberalisation" (line 94) - cannot agree with this as you show in Fig.1 that trade opennes is only the sum of exports and imports as percentages 68 of GDP

Reviewer 2 Report

1. The abstract is too long. Must be made succinct.

2. The introduction is too long. Recommendation is to reduce the introduction to a max of 5 paragraphs.

3. in line 75, the authors statement of strong correlation is not supported by evidence. For example, what is the correlation co-efficient?

4. The description of Figure 1 is not clear. Clarity must be provided.

5. The formulae for the ADF and the PP test are not needed as they can be found in any econometrics textbook.

6. The authors did not provide a priori expectations of the estimated coefficients in the long-term models. Would be good to provide them.

7. The lag variables in Table 5 are too many. They could use the "general to specific approach" to reduce them.

8. It would be better to show the p-values to ease of review or assessment.

9. Lines 368 to 376 are duplication from earlier part. Could be deleted.

10. The long-term co-efficient of the models were not provided. They must be shown.

11. What is the point of the Granger-Causality text in the current format of the paper? Is it needed?

12. Some of the paragraphs in the conclusion are not based on the results and findings of the paper. The authors must sanitised the conclusion

Reviewer 3 Report

The paper is primarily econometric in nature with application to the impact of exports, imports and trade openness on economic growth in Namibia. 

I address some suggestions and comments to the authors as follows:

-Consider shortening the introduction and transferring data and graphs on Namibia to the empirical part of the paper. The introduction should include the basic idea and problem of the study and a brief introduction to the methodology.

-As for the ARDL methodology, everything is done according to the standard ARDL procedure. The only complaint would perhaps be the excessive number of variables and parameters in Model 2, but since the variables are statistically significant, one can get by with this number (despite the fact that a larger number of parameters has been shown to increase the bias of the estimator, in this case OLS. This is shown in Model 2, say using the variable LABOUR as an example, so that the coefficient obtained is positive at the current time, in period t-1 it is negative, then in t-2 it is positive again but not statistically significant and at time t-3 it is positive but significant. The reason for these changes in the signs and values of the coefficients is precisely the high correlation between the variables, which causes the bias of the estimator and leads to the false conclusion that the variable is statistically significant when in fact it is not).

- In the fourth part, Results and Discussion, the discussion is not thematic but is integrated into the Conclusion, and the title of the fourth part should be changed to Results.

Round 2

Reviewer 2 Report

The authors could add the p-values in the table for the long-term model results.

Author Response

We take note of the comment you have made about the inclusion of the p-values of the long run equations. However, it is not normal practice to show the p-values of the long run equations given the fact that the estimated long run equations show spurious results whose significance cannot be interpreted. We are, therefore, of the conviction that adding the p-values of the long run equation will be a futile excerise, since they do not give us any additional information which is of use. We do not interpret the p-values of the results that are spurious.